

# The role of metacognition in promoting deep learning in MOOCs during COVID-19 pandemic

Marwa Yasien Helmy Elbyaly and  Abdellah Ibrahim Mohammed Elfeky

Faculty of Education, Najran University, Najran, Saudi Arabia
Faculty of Specific Education, Kafrelsheikh University, Kafrelsheikh, Egypt

## ABSTRACT

In many nations affected by the COVID-19 pandemic, the situation in higher education institutions has changed. During the pandemic, these institutions have introduced numerous e-solutions to continue the process of education. Besides, research has shown many benefits in the last years of MOOCs. Yet, to date there are few studies to explore some individual characteristics, such as learners' metacognitive skills, that might have an impact on learning outcomes in MOOCs. Furthermore, promotion of deep learning is a serious challenge for online courses including MOOCs. Therefore, the purpose of this research was to explore the role of metacognition in promoting deep learning in MOOCs during COVID-19 pandemic. The participants were students at the department of home economics who were all at the seventh academic level. Based on their scores on the metacognition awareness inventory (MAI), they were divided into two experimental groups, *i.e.* high metacognition students and low metacognition students. A three- aspect assessment card of deep learning namely connecting concepts, creating new concepts, and critical thinking was used to collect data. The results showed that MOOC was more effective in fostering the deep learning aspects of high metacognition skills, and deep learning as a whole. With regard to backward seeking and slow watching events, results showed significant differences in favor of high metacognition students (HMs). Nevertheless, there were no statistically significant differences between students in both groups regarding the pausing event.

## INTRODUCTION

The disease caused by the SARS-CoV-2 virus has been known as COVID-19. It first appeared in November, 2019 in the Chinese province of Wuhan and spread quickly around the world (*Dias & Lopes, 2020*). Education was one of the domains that has been tremendously affected by this pandemic that has been a real and immediate challenge to insecurity, health, unemployment and so on (*Surkhali & Garbuja, 2020*). To control this current outbreak, extensive steps have been introduced to minimize person-to-person transmission of COVID-19 (*Rothan & Byrareddy, 2020*). In education, and more specifically during the second semester of the academic year 2020, governments all over the world announced the closure of all schools and higher education institutions in an attempt to contain

Corresponding author
Marwa Yasien Helmy Elbyaly,
marwa.mohamed1@spe.kfs.edu.eg

COVID-19; Saudi Arabia was not an exception. Closure of educational institutions is a non-pharmaceutical measure used in many countries experiencing pandemics (*Doyle, 2020*). Thus, the alternative was to move from conventional to online learning in a scenario where learners are not allowed to go to educational institutions (*Basilaia & Kvavadze, 2020*).

Online courses were delivered for free and for anyone and the Internet connection became a common arena for large-scale instruction due to understanding of online learning and open access teaching movement (*Williams & Stafford, 2018*). Utilizing different educational technologies, most tertiary institutions nowadays, offer many opportunities for online learning (*Elfeky & Elbyaly, 2017*), such as Massive Open Online Courses (MOOCs). MOOCs are now generating considerable media attention and important interest from higher education institutions. MOOCs are a relatively modern online active learning phenomenon (*Yuan & Powell, 2013*), where active learning is the key to guarantee deep learning, *Biggs & Tang (2011)*, *Budd, Robinson & Kainz (2021)* and *Wu et al. (2021)*. In addition, worldwide enthusiasm for this pedagogical model that was believed to have the potential to revolutionize the educational delivery was stimulated by MOOCs (*Paton & Fluck, 2018*). Several researchers have pointed out that MOOCs have considerable potential for enhancing teaching and learning (*Adam, 2020*; *Chen et al., 2020*; *Doo & Tang, 2020*; *Ferguson & Clow, 2015*; *Kizilcec & Schneider, 2015*; *Mac Lochlainn & Nic Giolla Mhichíl, 2020*). However, the role of the metacognition skills in MOOCs are still not receiving the attention they deserve, despite the numerous studies that have been conducted to explore the impact of some individual characteristics on success in MOOCs (*Ashton & Davies, 2015*; *Milligan & Littlejohn, 2017*; *Prinsloo & Slade, 2019*).

Metacognition is often simplified as thinking about thinking or cognition about cognition (*Ku & Ho, 2010*). It addresses the conscious experience, self-regulation, and self-knowledge of one's cognitions or emotions (*Wagener, 2013*). It is related to the awareness and comprehension of a person regarding the cognitive phenomena (*Medina, Castleberry & Persky, 2017*). On the other part, deep learning requires activating the individual's awareness regarding the cognitive phenomena (*Biggs & Tang, 2011*; *Engel, Pallas & Lambert, 2017*). Metacognition variable is classified into two categories, high metacognition and low metacognition (*Redondo & López, 2018*). Educational research from the 21st century clearly demonstrates the necessity of the educational practices that help learners acquire the metacognitive abilities that they will need to succeed in the today's and future complicated and globalized society (*Howe & Wig, 2017*; *Howlett et al., 2021*; *Wafubwa & Csíkos, 2021*). On the other hand, deep learning includes creating new connections and concepts, integrating what students are learning with what they already know, in addition to critical thinking (*Filius et al., 2018*). Critical thinking is conceptualized as an operative higher order thinking example that can be accounted for because of validated and reliable tests, (*Miri & David, 2007*). Meanwhile, there is no doubt that metacognition is a core component of higher order thinking in various forms (*Ku & Ho, 2010*). Deep thinking learners can relate ideas and topic to prior experiences and knowledge (*Alt & Boniel-Nissim, 2018*). Besides, learners can develop a deeper approach to learning through the application of metacognition, resulting in greater academic achievement in courses

where expertise needs to be incorporated and applied (*Papinczak & Young, 2008*). In the context of MOOCs, deep learning may be a real challenge due to asynchronous written interaction and the lack of body language and visual cues (*Filius et al., 2018*; *Henderikx & Kreijns, 2019*). In other words, the sense of community and interaction can be seen as deep learning prerequisites (*Ertmer et al., 2007*). MOOCs can be an online learning form of higher education that has a strong potential to enhance deep learning and as long as learners do not see each other, such an interaction will mostly written and asynchronous and could have consequences for choosing an approach of deep learning (*Filius et al., 2018*). Video lectures are also one main part of the MOOC course design where the learning platforms store data of web log including student experiences with the course material *e.g.*, video interaction events (*Mubarak, Cao & Ahmed, 2021*). The event of video interaction is cognitive engagement including pausing, backward seeking, and video slow watching (*Li & Baker, 2018*). Experimental proofs from early research on interactive educational videos in online learning shows that allowing the learners the chance to interact with videos through pausing, backward seeking, and slow watching, significantly improves learning (*Tang & Xing, 2018*; *Xing, 2019*; *Zhang & Zhou, 2006*). Moreover, a MOOC participant leaves behind an easily accessible log of behaviors, such as information about every time he begins rewinds or pauses a video. Despite these attributes of a MOOC, much of the previous research does not directly discuss the actual cognitive processes underlying events of video interaction (*Li & Baker, 2018*), and does not investigate the relationship between metacognition and video interaction events in MOOC. Henceforth, adaptation of the presented information to the learner's cognitive processing needs, such events may let him control the density of the presented information, speed, and order (*Brinton & Buccapatnam, 2015*; *Li & Baker, 2018*; *Zhang & Skryabin, 2016*).

Henceforth, this research aims to investigate the extent to which metacognition variable can promote the intended outcomes of deep learning like connecting concepts, creating new concepts, and critical thinking, (*Filius et al., 2018*). In addition, it aims to measure the extent to which students' pausing, backward seeking, and video slow watching can be used to infer the relationship between metacognition and video interaction events in MOOCs during COVID-19 pandemic. In short, it aims to answer these questions:

RQ1: To what extent does the MOOC promote deep learning namely, connecting concepts, creating new concepts, and critical thinking of students of high metacognition and low metacognition?

RQ2: Does the learners' metacognition, whether high or low, impact events of video interaction like slow watching, backward seeking, and pausing in MOOCs?

## LITERATURE REVIEW

### MOOCs

MOOCs have emerged as a popular mechanism for individuals to acquire acquiring new skills and knowledge (*Milligan & Littlejohn, 2017*). Hence, a primary goal of MOOCs is to provide people an opportunity to learn (*Kizilcec & Pérez-Sanagustín, 2016*). MOOCs are unlike most other types of online learning in higher education. They are free and are

funded by top-tier institutions that offer them an air of prestige that has never been achieved before by online courses (*Evans & Baker, 2016*). Meanwhile, they encourage learners to study when and where they choose. There is an improvement in the autonomy of learners attending a MOOC compared to those attending a conventional course (*Jansen & Van Leeuwen, 2017*). MOOCs are rapidly a growing method of educational provision (*Hone & Said, 2016*). They can be seen as an expansion of current online education approaches, in terms of scalability and open access to courses (*Yuan & Powell, 2013*). Their course structures consist of auto-graded quizzes, online discussion forums, and lecture videos (*Lee & Watson, 2020*). In other words, they are built as an alternative to most practices of traditional online learning that deliver content through single or centralized platform (*Joksimović et al., 2018*). Therefore, in the coming years, MOOCs are expected to be playing a key role in the learning of undergraduate students.

## Deep learning

Surface learning and deep learning approaches are two main forms of learning by which learners can learn (*Filius et al., 2018*). Deep learning is a process of learning advocated by the theory of constructivist learning that occurs through students' social negotiation, collaboration, and reflection on their own practices of learning advocated by theory of constructivist learning (*Lee & Baek, 2012*). Besides, it is an approach of complex personal development involving the change of learning habits, epistemological beliefs and perceptions. It focuses on the underlying meanings, main ideas, themes and principles. It also stresses the importance of refining ideas, applying knowledge and utilizing evidence across contexts (*Biggs & Tang, 2011*; *Donnison & Penn-Edwards, 2012*; *Wingate, 2007*). In contrast, surface learning is a passive treatment of information, employs low-level metacognition, and lacks thinking (*Lee & Baek, 2012*). In addition, it treats the course as routinely memorizing facts and carrying out procedures. It focuses on unrelated bits of knowledge and lower requirements of syllabus (*Donnison & Penn-Edwards, 2012*; *Entwistle & Peterson, 2004*).

Deep learning requires learners to relate ideas and topics to prior experiences and knowledge as an activity of constructivist education, which refers to the idea that skills and content should be understood within the student's prior knowledge framework (*Alt, 2018*; *Alt & Boniel-Nissim, 2018*). On the other side, surface learning, which is confined to memorizing facts and rote learning, requires students to memorize or replicate the learning material for a test (*Filius et al., 2018*; *Price, 2014*). With the latter, only the basics of the learning material are learned (*Rozgonjuk & Saal, 2018*). Nevertheless, deep learning is made up of three main aspects namely, creating new concepts, connecting concepts, and critical thinking (*Filius et al., 2018*). Further, deep learning leads to higher academic success and performance (*Karaman & Demirci, 2019*; *Uludag & Uludag, 2017*). In professional environments where online tools are to be utilized, deep learners can be efficiently supported (*Lee & Baek, 2012*; *Li & Xing, 2021*) although it may be a challenge in the context of MOOCs as mentioned earlier.

## Metacognition

Metacognition is involved in most learning situations (*Wagener, 2013*) because it refers to a person's cognition and knowledge regarding the cognitive phenomena (*Medina, Castleberry & Persky, 2017*). It enables the student to be more aware of the achieved progress (*Tops & Callens, 2014*). In other words, metacognition refers to one's own thoughts and cognitions (*Driessen, 2014*). It is the highest form of one's intellectual capacity (*Paliokas, 2009*). Knowledge about the cognitive tasks, strategic knowledge, and self-knowledge are the main substructures of metacognition (*Polegato, 2014*). During the process of learning, metacognition directs the students' learning strategies (*Medina, Castleberry & Persky, 2017*) where strategies of metacognition represent an important variable (*Halpern, 1998*). Aspects of metacognitive interact with a variety of external and internal factors such as socio-economic status, motivation, and type of instruction (*Medina, Castleberry & Persky, 2017*). The Metacognitive Awareness Inventory (MAI) developed by *Schraw & Dennison (1994)* usually measures learners' metacognitive skills. It follows a common model of two components, *i.e.,* Regulation of Cognition and Knowledge of Cognition (*Mäkipää, Kallio & Hotulainen, 2021*). Regulation of Cognition expresses the students' need to modify to students' modifying the progress of their cognitive activity and control of their own cognitive processing (*Cleary & Kitsantas, 2017*) while Knowledge of Cognition refers to what learners know about their own cognition or about cognition in general. It involves procedural, conditional, and declarative knowledge (*Mäkipää, Kallio & Hotulainen, 2021*). In brief, interest in the metacognition role has been steadily rising in most education forms (*Meijer et al., 2013*). Several researches have put forth that metacognition is a milestone variable to estimating the learning performance (*Baş & Sağırlı, 2017*; *Veenman, 2006*). Therefore, the present study aims at investigating the role of metacognition in the promoting of deep learning in MOOCs.

## METHODOLOGY

### Participants

Participants in the present study were (59) students at the department of home economics at Najran University. They were all at their seventh level and were all enrolled in "Research Paper Writing" course that was provided *via* Coursera platform. The quantitative MAI whose reliability coefficient was confirmed by *Schraw & Dennison (1994)* and validated by *Sperling & Howard (2004)* was used to assess the level of metacognitive awareness of each student. Table 1 reveals that the mean score and standard deviation of students in the first group was ($M = 198.63$ & SD $= 9.74$), while it was ($M = 141.26$ & SD $= 11.42$) for the second group. That is, out of a total of 260 quantitative MAI points, metacognition was graded into two groups, high metacognition (HM $\geq$ 65 per cent) and low metacognition (LM $<$ 65 per cent) in line with *Aydın & Coşkun (2011)* and *Redondo & López (2018)*. In other words, based on their scores on the MAI instrument, participants were divided into two experimental groups, the first group consisted of (27) high metacognition female students, while the second one involved (32) low metacognition female students. Their average age was 21 years and the standard deviation was 1.76. Before proceeding learning,

**Table 1  Differences between participants' metacognition levels.**

|  | Group | $n$ | M | SD |
|---|---|---|---|---|
| Metacognition | HMs[*] | 27 | 198.63 | 9.74 |
|  | LMs[**] | 32 | 141.26 | 11.42 |

Notes.

   HMs[*] are high metacognition students; LMs[**] are low metacognition students.

**Table 2  Differences between the two groups regarding participants' previous deep learning as a whole on the Pre-application of the assessment card.**

|  | Sum of squares | DF | Mean of squares | $F$. ratio | Sig. |
|---|---|---|---|---|---|
| Between groups | 2.23 | 1 | 2.23 | 1.08 | 0.541 |
| Within groups | 887.6 | 57 | 15.74 |  |  |
| Total | **889.83** | **58** |  |  |  |

**Table 3  Differences between the two groups in terms of previous findings of "Computer in Teaching" course.**

| Course |  | Sum of squares | Mean square | $F$. ratio | Sig. |
|---|---|---|---|---|---|
| Computer in teaching | Between groups | 13.62 | 13.62 | 1.77 | 0.583 |
|  | Within groups | 417.49 | 7.14 |  |  |
|  | Total | 431.11 |  |  |  |

all participants were informed of the research aim and signed consent forms. They were given the opportunity to not participate and withdraw without penalty. The researchers confirmed that, for research involving human subjects, they have met ethical guidelines.

In addition, homogeneity of learners' previous deep learning as a whole were checked using ANOVA after the pre-application of the assessment card. Results in Table 2 show that F. ratio (1.08) was not significant at ($\alpha = 0.541 > 0.05$). In other words, there were no statistically significant differences in learners' deep learning as a whole on the pre-application of the assessment card for both groups. One interesting explanation also for this, lies in their previous enrollment and success in the "Computer in Teaching" course that developed their technology skills. As shown in Table 3, F. ratios (1.77) were also insignificant ($\alpha = 0.583 > 0.05$) and so we can claim that all participants' technology skills were also homogeneous in the course of the "Research Paper Writing" course.

## Study procedure

It is important to bear in mind that the use of technology in educational settings needs to be based on the dominant theories and methods of education (*Elfeky, Masadeh & Elbyaly, 2020*; *Patten & Sánchez, 2006*). Consequently, this is applied to MOOCs as one type of educational use of technology (*Elfeky & Elbyaly, 2016*; *Elfeky & Elbyaly, 2021*). MOOCs are based on the theory of connectivism learning (*Siemens, 2014*), which is a new theory utilized *via* social learning experiences to explain learning in the digital age, with a focus on students making connections to skills and knowledge (*Paton & Fluck,*

*2018*). The current research is a three-phase method. In the first phase, metacognition of students was evaluated utilizing the Metacognitive Awareness Inventory (MAI) instrument expanded by *Schraw & Dennison (1994)*. It is a questionnaire of 52 questions of the five point-Likert scale. Responses were ranging from 'strongly agree' to 'strongly disagree'. In the second phase, the "Research Paper Writing" course was offered to participants *via* Coursera platform (http://www.coursera.org). In the third phase, students' outcomes of intended deep learning in MOOC were assessed using an assessment card and data of video interaction log collected *via* the Coursera platform.

Participants were invited to take part in a MOOC by one of the research team member through Zoom platform. A 15-minute MOOC orientation lecture was given with a clarification of how MOOC could be utilized as a resource for research paper writing. Help was offered when needed through the Zoom platform to complete the MOOC sign-up operation. Participation was voluntary, free, promoted and encouraged through the research team, *i.e.,* course teaching team. The MOOC lasted for six weeks from March to May 2020. Each week, two blocks on related subjects were introduced. Each block comprised of 45 min of devoted to studying background materials and consisted of two phases. The first 30 min were allotted to assignments or tasks, and the other 25 min were assigned for video watching. More specifically, the "Research Paper Writing" course was delivered *via* the Coursera platform that aimed to allow participants to practice what they need by having them to finally write a research paper. Each week covered one or more topics about the research paper writing, such as, selecting an academic topic, formulating an appropriate research question, creating an outline, and looking for source material and researching. In addition, participants were to create a bibliography of annotated, write several paragraphs including the introductory paragraph, work cited page, and finally carefully review and edit the research paper. Once research papers, as an assignment, were submitted, participant students' deep learning was to be assessed using the assigned assessment card. Meanwhile, data of video interaction log were collected *via* the Coursera platform in order to infer relationship between metacognition and events of video interaction including data about learners' every video player clicks mainly slow watching, pausing, and backward seeking. An event of pausing was defined while watching the video as the learner could stop the video lecture by clicking the pause button. An event of backward seeking was also defined as moving the video head of play to a new position before the old position by the learner, *e.g.,* changing the video head of play from marker 15:20 to marker 12:41. In addition, an event of slow watching was known as the learner's changing video playing speed to slower one than it was before changing.

Besides, *via* an assessment card of deep learning, the intended outcomes of deep learning were assessed at the end of the course. A team of three professors rated all the research papers. Through discussion, the main differences in the evaluation were overcome. For scoring, signs agreed upon were utilized. Data of assessment card was used to identify utility of metacognition in promoting deep learning in MOOCs.

### Instruments of data collection

In order to describe students' behaviors associated with deep learning, an assessment card of three main aspects namely connecting concepts, creating new concepts, and critical thinking was developed. Items of these three main aspects constituted the deep learning operationalization based on the Approaches to Study Inventory (ASI) of *Entwistle & Ramsden (1983)* and the Study Process Questionnaire (SPQ) of *Biggs (1987)*. To validate the content of the prepared assessment card, it was presented to a set of arbitrators, who were all experts in the fields of home economics, curricula and instruction methods, and educational technology. The total number of the assessment card's items was 19. Critical thinking aspect consisted of seven items, the aspect of connecting concepts also involved seven items while only five items constituted the creating new concepts aspect. Participants' responses to these items ranged from 1 = Strongly disagree to 5 + Strongly agree on a five-point Likert Scale (See Appendix A). Furthermore, using Cronbach Alpha to ascertain the card's reliability, the card's internal reliability was 0.89 (critical thinking: 0.877, connecting concepts: 0.849, and creating new concepts: 0.854). To ensure the inter-rater reliability of the evaluation results, an independent professor was requested to analyze and check the papers of approximately 10% of the whole research papers. Agreement percentage of all raters was approximately 92%.

### Data analysis

Quantitative and qualitative data were taken into consideration. More specifically, results of the assessment card were taken as a beginning point for the analysis to explore the role of metacognition in promoting deep learning. Data of video interaction log were also accounted for in order to infer the relationship among video interaction events and metacognition in MOOCs. Also the independent sample $t$-test was utilized, and a significance level of $p < 0.05$ was adopted for the research.

### Ethical statement

Approval was received from the Deanship of Scientific Research review board at Najran University (10/918/1442/137). The procedures used in this study adhere to the tenets of the Helsinki Declaration.

## RESULTS

### Usefulness of metacognition in promoting learners' ability in critical thinking in MOOCs

Results related to critical thinking aspect presented in Table 2 show that there are significant differences between learners of high metacognition skills (HMs) and their peers of low metacognition skills (LMs) with regard to their critical thinking skills ($P = .000 < .05$). Mean scores of both groups obviously indicate that critical thinking of learners in the HM group was better than the critical thinking of their peers in the LM group in MOOCs. To say it in a different way, MOOC was more effective in promoting the critical thinking of high metacognition students. Eta Square ($\eta2$) was utilized to determine how much the learners' ability in the HMs group to think critically in the HMs group was boosted in order

to ascertain this finding. The estimated value ($\eta2 = 0.305$) confirms that the additional value of MOOC utilization in enhancing the ability of high metacognition learners to think critically was more effective than in enhancing the ability of low metacognition students. More specifically, results show that MOOC was more effective in the enhancement of HMs' ability in testing the effect of the independent variable on the dependent one, identifying the study questions to be answered, and formulating probable answers that can be tested for every question. In addition, it was more effective in writing the study null and alternative hypotheses, enhancing high metacognition participants' ability to distinguish between hypotheses that can be tested descriptively or quantitatively, concluding results in a shorter time, and including the answer of a previous study in the topic of the chosen study.

## Usefulness of metacognition in promoting learners' ability to connect concepts in MOOCs

Table 4 presents the findings related to the usefulness of metacognition in promoting learners' ability to connect concepts, *i.e.,* connect new knowledge with what students' already knew. Results indicate statistically significant differences in connecting concepts between learners in the HMs and LMs groups ($P = .000 < .05$). That is, the ability of HM students to connect concepts as reflected on the assessment card was much better than that of the LM students in MOOCs. In other words, MOOC enhancement of the ability of high metacognition learners to connect concepts was much better than the enhancement of the ability of peers with low metacognition. In order to ascertain this finding, Eta Square was utilized to determine how much the learners' ability to connect concepts in the HMs group was boosted. The estimated value ($\eta2 = 0.297$) confirms that the additional value of MOOC utilization was more effective in enhancing the ability of high metacognition learners to connect concepts than in enhancing the learners of low metacognition ability. More specifically, results show that MOOC was more effective in the enhancement of HMs' ability in considering the criteria for formulating a good research title, writing a key question that the study will answer, and identifying the study population. Besides, it was more effective in documenting references and resources; identifying independent, dependent and persistent variable; describing the sampling technique and type; and identifying topic related laws, principles or theories.

## Usefulness of metacognition in promoting the learners' ability to create new concepts in MOOCs

Results related to the usefulness of metacognition in creating new concepts are presented in Table 4. They indicate statistically significant differences in creating new concepts between students in the HMs and LMs groups ($P = .000 < .05$). That is, HMs' ability to create new concepts as reflected on the assessment card was much better than that of LMs' ability in MOOCs. In other words, MOOC enhanced the ability of high metacognition learners to create new concepts much better than the ability of their peers with low metacognition. The Eta Square was utilized to determine how much the learners' ability to create new concepts in the HMs group was boosted in order to ascertain this finding. The estimated value ($\eta2 = 0.341$) confirms that the additional value of MOOC utilization was more effective in enhancing the ability of high metacognition learners to create new concepts than the ability

**Table 4** Differences between participants' deep learning promotion (creating new concepts, connecting concepts, and critical thinking) in HMs and LMs groups.

|  | Groups | *n* | M | SD | *t* | *p* |
|---|---|---|---|---|---|---|
| Critical thinking | HMs[*] | 27 | 4.01 | .341 | 6.12 | .000[***] |
|  | LMs[**] | 32 | 3.36 | .450 |  |  |
| Connecting concepts | HMs | 27 | 4.24 | .436 | 4.77 | .000[*] |
|  | LMs | 32 | 3.69 | .440 |  |  |
| Creating new concepts | HMs | 27 | 4.27 | .389 | 4.83 | .000[*] |
|  | LMs | 32 | 3.78 | .391 |  |  |
| Deep learning as a whole | HMs | 27 | 4.17 | .296 | 7.43 | .000[*] |
|  | LMs | 32 | 3.61 | .283 |  |  |

Notes.

HMs[*] are high metacognition students. LMs[**] are low metacognition students $p^{***} < .05$.

of peers with low metacognition. In brief, results show that MOOC was more effective in the enhancement of HMs' ability to formulate the terms associated with the result and its causes or phenomena and their conditions, describe the proposed experimental design, identify the data collection techniques and tools, identify topic related data or results, and process data or results.

## Usefulness of metacognition in promoting deep learning as a whole in MOOCs

The findings as shown in Table 4 present statistically significant differences in deep learning as a whole between students in the HMs and LMs groups ($P = .000 < .05$). In other words, HMs' deep learning was much better than that of LMs' deep learning. The result given is not a sudden, since it is based on the previous results. To determine the amount of improvement in participants' deep learning as a whole, the Eta Square was utilized in order to ascertain this finding. The estimated value ($\eta2 = 0.372$) confirms that the additional value of MOOC utilization was more effective in enhancing the ability of high metacognition learners' deep learning as a whole than the deep learning of peers with low metacognition. Besides, this result confirms that MOOC was more effective in fostering the deep learning as a whole of high metacognition learners.

## Usefulness of learners' metacognition in video interaction events in MOOCs

Results related to video interaction events presented in Table 5 shows that there were no statistically significant differences between participants in both groups with regard to pausing ($P = .883 > .05$). On the contrary, there were statistically significant differences with regard to backward seeking in both groups ($P = .038 < .05$). Mean scores of both groups obviously indicate that participants' backward seeking in HM group was greater than the backward seeking of peers in the LM group in MOOCs. Besides, there were statistically significant differences between both groups with regard to the slow watching in ($P = .032 < .05$). Mean scores of both groups obviously indicate that HM participants' slow watching was greater than the slow watching of peers in the LM group in MOOCs.

**Table 5  Differences between participants' video interaction events in HM and LM groups in MOOC.**

| Video interaction events | Groups | M | SD | t | p |
|---|---|---|---|---|---|
| Pausing | HMs | 11.93 | 11.201 | .186 | .883 |
| | LMs | 11.38 | 11.401 | | |
| Backward seeking | HMs | 12.30 | 5.143 | 2.085 | .038[*] |
| | LMs | 9.81 | 4.004 | | |
| Slow watching | HMs | 10.19 | 3.574 | 2.262 | .032[*] |
| | LMs | 8.34 | 2.671 | | |

**Notes.**
$p^* < .05$.

That is, it can be inferred that high metacognition of participants was more effective in backward seeking and slow watching in video lecture in MOOCs.

## DISCUSSION

Best understanding of how learning outcomes are related to metacognition is important for identifying participants' behavior in internet-enabled learning. The overarching aim of the current research was to disclose the role of high and low metacognition in promoting deep learning represented in creating new concepts, connecting concepts, and critical thinking. It also aimed to measure slow watching, backward seeking, and pausing of videos in order to infer the relationship among video interaction events and metacognition in MOOCs during COVID-19 Pandemic. Major results of this study can be explained in light of deep learning aspects. First, results suggest that MOOC was more effective on high metacognition participants than on low metacognition ones in promoting critical thinking. Such a result emphasizes what *Medina, Castleberry & Persky (2017)* has claimed regarding the ability of metacognition to improve thinking and learning. In addition, metacognition constitutes an essential part of cognitive development cognitively to make critical thinking possible (*Kuhn, 1999*). It has an important path to critical thinking (*Magno, 2010*). Besides, this result corroborates the findings of *Arslan (2018)* and *Naimnule & Corebima (2018)*, which also found that there was a relationship among the skills of critical thinking and metacognitive where critical thinking positively predicted metacognition. Furthermore, the findings of this study indicated that MOOC was more influential on high metacognition learners in enhancing connecting concepts, *i.e.,* connecting new knowledge with what they already know. Such a result confirms that MOOC could provide participants with support to appropriately build on their previous ideas and on how to coherently construct their new ones.

One more interesting thing is the fact that the findings of this research confirm the role of using MOOC in promoting students' abilities in creating new concepts, particularly, high metacognition ones. Hence, participants in the present study, as *Redondo & López (2018)* mention, were able to innovate, exercise their intellectual capacities, and approach novel processes. Another important fact these results foster, is the fact that deep learning of high metacognition learners was found much better than that of peers of low

metacognition. Consequently, it can be said that MOOC is usually more effective with high metacognition learners in fostering their deep learning as a whole. This conclusion is, to a large extent, in line with *Tsai & Lin (2018)* that enhancing metacognition of learners can lead to continuance to learn with MOOCs and increased online learning interest. Furthermore, findings of this study are in harmony with *Barak & Watted (2016)* that MOOC success search should include a better learner characteristics' understanding from different disciplines. Therefore, high metacognition is highly needed once lecturers seek to promote students' deep learning in MOOCs. In short, it can be said that findings of this study can contribute to study and practice around learner characteristics and learning outcome in the context of MOOCs during the Corona Virus pandemic and the future, in general. In particular, these findings can strengthen our notion that intended deep learning outcome might be connected to learners' metacognitive skills in MOOCs.

As for the cognitive processes that underlie video interaction events, findings revealed that there was a relatively significant relationship between backward seeking event and high metacognition in MOOC. This result, to a large extent, corroborates the findings of *Li & Baker (2018)* regarding the significant relationship among students' course grades and backward seeking. That is, the event of backward seeking is positively linked to utilizing cognitive strategies and investing mental effort. High metacognitive skills, on the other part, help participants to understand what they know and do not know and consequently help them get the missing information that is called as self-directed or self-regulated learning (*Medina, Castleberry & Persky, 2017*). Many MOOC researches shows that better quiz results are predicted by backward seeking events (*Brinton & Buccapatnam, 2015*; *Li & Baker, 2016*). Similar to the previous theme, results of this study also indicated a relatively significant relationship between slow watching event and high metacognition in MOOC. Such a result highlights what has been claimed by *Sinha & Jermann (2014)* about the positive association between in-video persistence, slow watching event, and persistence of the course. Once again, this result supports the findings of *Li & Baker (2018)* with regard to the fact that, for all-rounders, slow watching was indicative of greater course grades. In other words, high metacognitive skills allow learners to be more conscious of the progress made (*Tops & Callens, 2014*). However, findings found that there were no statistically significant differences between the two groups with regard to pauses. Findings such as these can be attributed to certain reasons or facts like, for example, the fact that pausing event can be seen as an indication bout the cognitive load increase (*Van Merrienboer & Sweller, 2005*). Findings may also prove what *Li & Baker (2018)* claim that for reasons unrelated to learning, learners may pause, like taking a break to do something else.

## LIMITATIONS AND FUTURE DIRECTIONS

This research had several limitations. First, exploration of the role of metacognition in promoting deep learning in MOOCs was the focus of the present study. Second, implementation of the current study was limited to a sample of female students of home economics major. Therefore, researchers are invited to carry out similar researches in other environments to explore the role of metacognition in MOOCs where male and female

learners study together. Third, female learners in this study were tertiary students, so our results can not be compared with other age groups as *Elfeky (2017)* states. Fourthly, data analyzed were collected from one public tertiary institutions so generalizability of results can not be done. Therefore, researchers of future studies are called to utilize data from different institutions in other countries (*Elbyaly & Elfeky, 2021*), they are called to reveal the impact of utilizing the Big Data analytics in MOOCS to promote deep learning for learners with low metacognition.

## CONCLUSION

The present study is a three-phase method study where participants' metacognition was evaluated in the first phase. The second phase involved the delivery of the ''Research Paper Writing'' to participants *via* the Coursera platform. In last phase, the intended deep learning outcomes in MOOCs were identified, and data of video interaction log were collected *via* Coursera platform. The overarching aim of the current research was to disclose the role of metacognition, high and low in promoting various aspects of deep learning. *i.e.,* creating new concepts; connecting concepts; and critical thinking. It also targeted measuring slow watching, backward seeking, and pausing of videos in order to infer the relationship among video interaction events and metacognition in MOOCs during COVID-19 Pandemic. Results proved that high metacognition could promote learners' critical thinking, connecting concepts, creating new concepts, and deep learning as a whole in MOOCs. In other words, metacognitive skills matter and supporting these skills can help to also promote students' deep learning in MOOCs. They also showed statistically significant differences with regard to backward seeking and slow watching events in favor of HMs, while no statistically significant differences with regard to pausing event were noticed.

### Funding
This work was supported by the Deanship of Scientific Research at Najran University for funding this work through grant research code NU/SEHRC/10/918. The funders had no role in study design, data collection and analysis, decision to publish, or preparation of the manuscript.

### Grant Disclosures
The following grant information was disclosed by the authors:
Deanship of Scientific Research at Najran University: NU/SEHRC/10/918.

### Competing Interests
The authors declare there are no competing interests.

## Author Contributions

- Marwa Yasien Helmy Elbyaly conceived and designed the experiments, performed the experiments, performed the computation work, authored or reviewed drafts of the article, and approved the final draft.
- Abdellah Ibrahim Mohammed Elfeky conceived and designed the experiments, performed the experiments, analyzed the data, prepared figures and/or tables, and approved the final draft.

## Data Availability

The raw data is available in the Supplementary File.

## Supplemental Information

Supplemental information for this article can be found online at http://dx.doi.org/10.7717/peerj-cs.945#supplemental-information.

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
