# Peer review of "The role of metacognition in promoting deep learning in MOOCs during COVID-19 pandemic"

_PeerJ Computer Science, doi:10.7717/peerj-cs.945_

## Round 0.1 · original submission · Major Revisions

The paper needs further major revisions - the authors should clarify more about the dataset used and its features. Also the training ad testing model should be explained in a systematic way.

The authors should follow the comments from the reviewers precisely and clearly.

·

Basic reporting

The paper aims at addressing role of metacognition in promoting deep learning via MOOCs. The work targets to answer two research questions. The work reported though not novel is acceptable contribution as the attempt is considerable.

Experimental design

Three aspects Critical thinking, Connecting concepts and Creating new concepts are considered as evaluation strategy. The questionnaire enclosed along with manuscript is considered for review.

In my view, A detailed analysis w.r.t Anova would have been better instead of just stating the outcome. Some more w.r.t analysis and key findings would add value. Authors can address these in revision.

Validity of the findings

Computation enclosed in excel file support for validation. Conclusion is relatable to discussion done in the paper. The approach though not novel is acceptable attempt.

Additional comments

A more comparative evaluation is preferred as present version is limited.

Usually, analysis is expected but analysis part is very limited

These two points may be considered

Reviewer 2 ·

Basic reporting

its an attempt to explore the Role of Metacognition in Promoting Deep Learning in MOOCs during COVID-19 Pandemic. deep learning is not elevated in detail. data set taken for experimentation is not mentioned in detail.

Experimental design

needs more explanation about dataset taken and its features. how they have trained the model and how they tested. utilization of deep learning model to be discussed. Results generated from the proposed idea are limited

Validity of the findings

somewhat okay but dataset size and features to be enhanced. model should be more scalable with more variety of data and with higher dimensions are able to process in dynamic platform.

Additional comments

more investigation to be done on idea(methodology), dataset taken and projecting the role of deep learning in MOOCs

Reviewer 3 ·

Basic reporting

As per my observation, the studies were done on covid -19 and are related to the title given. Participants were students at the department of home economics who were all at the seventh academic level. Based on their scores on the metacognition awareness inventory, they were divided into two experimental
groups, i.e high metacognition students and low metacognition students done with a small sample.
if it is taken for a large sample will the same results will be appeared?

Experimental design

As per the Author's view 203 out of 260 quantitative MAI points, metacognition was graded into two groups, high 204 metacognition (HM ≥ 65 percent) and low metacognition (LM < 65 percent) in line with Aydın205 and Coşkun (2011); Redondo and López (2018).
In other words, based on their scores on the 206 MAI instrument, participants were divided into two experimental groups.
There is a scope for comparative study. is it possible to incorporate?

Validity of the findings

is it justifiable that the authors noted that all participants have studied and passed the course's prerequisite 14 courses, mainly Research Methodology, Teaching & Statistics Principles, and Principles of
215 statistics. Performance can be upgraded

Additional comments

Check the Grammar again

---

## Round 0.2 · accepted · Accept

The paper can be accepted without further revisions.

·

Basic reporting

All responses are addressed in the revised version of the manuscript.

Experimental design

Investigation is carried and method described are sufficient w.r.t revised version of manuscript.

Validity of the findings

Findings are validated with proper data.

Additional comments

All review comments are answered and incorporated appropriately.

Reviewer 2 ·

Basic reporting

manuscript is well organized and explored on important point required in current context

Experimental design

methodology given well and implemented too to obtain specific outcomes

Validity of the findings

findings obtained from the proposed method are valid may be improved in the performance on various datasets/various data sizes

Additional comments

manuscript will be so worthy if mentioned Limitations are reduced and future directions are implemented further

Reviewer 3 ·

Basic reporting

Yes, the revision version is fine.

Experimental design

The results enclosed in revision are fine

Validity of the findings

Supporting data is provided for validation.

Additional comments

The revision version is acceptable.